# Effect of Different Fine Aggregate Characteristics on Fracture Toughness and Microstructure of Sand Concrete

**DOI:** 10.3390/ma16052080

**Published:** 2023-03-03

**Authors:** Zhihua Sun, Jin Xiong, Shubo Cao, Jianxiong Zhu, Xuzhi Jia, Zhigang Hu, Kaiping Liu

**Affiliations:** 1School of Materials Science and Engineering, Chang’an University, Xi’an 710061, China; 2Shaanxi Hongcheng Mining Technology Corporation Limited, Xi’an 710001, China; 3Beijing Building Research Institute Corporation Limited of CSCEC, Beijing 100076, China

**Keywords:** fine aggregate, sand concrete, fracture toughness, microstructure

## Abstract

The fracture toughness of sand concrete is affected by aggregate characteristics. In order to study the possibility of exploiting tailings sand, available in large quantities in sand concrete, and find an approach to improve the toughness of sand concrete by selecting appropriate fine aggregate. Three distinct fine aggregates have been used. After characterizing the fine aggregate used, the mechanical properties were tested to characterize the toughness of sand concrete, the box-counting fractal dimensions were calculated to analyze the roughness of fracture surfaces, and the microstructure was tested to observe the path and width of microcracks and hydration products in sand concrete. The results show that the mineral composition of fine aggregates is close, but their fineness modulus, fine aggregate angularity (FAA) and gradation vary considerably; FAA has a significant impact on the fracture toughness of sand concrete. The higher the FAA value, the more resistant it is to crack expansion; with the FAA values of from 32 s to 44 s, the microcrack width in sand concrete was reduced from 0.25 um to 0.14 um; The fracture toughness and microstructure of sand concrete are also related to the gradation of fine aggregates, the better gradation can improve the performance of the interfacial transition zone (ITZ). The hydration products in the ITZ are also different because more reasonable gradation of aggregates reduces the voids between the fine aggregates and the cement paste and restrains the full growth of crystals. These results demonstrate that sand concrete has promising applications in the field of construction engineering.

## 1. Introduction

Sand concrete is a new type and special composite material [1,2,3], which has attracted widespread attention due to its excellent mechanical properties and low self-weight relatively. These excellent properties are achieved by means of using fine aggregates and reactive fillers to enhance the close packing between particles [4,5]. More hydration products are formed at the interface in sand concrete, directly contributing to the densification of the grout-aggregate interface. Thus, the mechanical properties of sand concrete are increased effectively. Fracture toughness has been considered one of the important factors affecting the application of sand concrete, which reflects the ability to resist crack propagation during deformation and fracture. The fracture toughness of sand concrete affects its further application in the engineering field. Therefore, it is crucial to explore the fracture toughness of sand concrete [6].

The fracture toughness of concrete is usually improved by adding supplementary cementitious materials, inert fillers, reactive fillers and fibers or by improving curing methods. For instance, Madani Bederina et al. [7] studied the fracture toughness of sand concrete by adding limestone fillers and the results showed that the best values of flexural strength were obtained by using a limestone content which was about 5%. Chaohua Jiang et al. [8] compared the effects of ground-granulated blast-furnace slag and fly ash on the fracture toughness of sand concrete; the results indicated that the fracture toughness of sand concrete with ground-granulated blast-furnace slag increased 15.5% compared with sand concrete with fly ash. Huajian Li et al. [9] studied the fracture toughness of sand concrete with active fillers as supplementary cementitious materials; the results showed that the 56 days flexural strength was best when the cement was replaced 20% as a supplementary cementitious material and slightly increased by 7.2%. Weichen Tian et al. [10] analyzed the fracture toughness of sand concrete with dolomite powder as fine aggregate by different curing methods; the results showed that the fracture toughness was comparable by using low-energy ohmic heating curing and traditional high-temperature steam curing. Belhadj et al. [11,12,13,14,15] added barley straws and other renewable materials into sand concrete as toughening phases; the results showed that barley straw used alone or mix not only improved the toughness of sand concrete properly but also retained the lightweight properties. And there are many studies on the use of mineral admixtures and their effect on the performance of concrete. Zhu [16] used iron tailings (particle size greater than 0.075 mm) to prepare sand concrete instead of fine aggregate, and the results showed that the addition of iron tailings can produce high performance sand concrete; Ivan Francklin Junior [17] used quartzite tailings to produce high-performance concrete; Siddique [18] think waste foundry sand could be very conveniently used in making concrete with good quality. Zhang [19] studied the utilization of copper tailing as micronized sand to prepare high-performance concrete. Yugen Li [20] studied the durability of sand concrete prepared by replacing river sand with aeolian sand. However, there is little in detail literature on the effects of the aggregate property on the fracture toughness of sand concrete.

The crack propagation directly affects the fracture energy and fracture toughness. The fracture process of sand concrete with different aggregates is shown in Figure 1. When the aggregate is round and contains coarse particles, it is easy to cause stress concentration and penetration cracks during crack propagation [21,22] (Figure 1a). When the aggregates are very fine and have angular shapes, the cracks can bypass the aggregate and result in a longer propagation path (Figure 1b). The crack paths of sand concrete with fine aggregates are more tortuous than sand concrete with coarse aggregates, which results in higher energy consumption during crack expansion in sand concrete with fine aggregates. Since the crack formation in concrete requires debonding between the surface of aggregate particles and the cement paste, the cracks propagate gradually with the increase of bending stress. The cracks are constantly cycling from particle surface debonding to deflection or branch during propagation. This further lengthens the path of crack propagation and consumes more fracture energy; thus, the fracture toughness of sand concrete with fine aggregates is improved. The crack propagation is also influenced by the aggregate gradation. Consequently, the characteristics of fine aggregates have an important effect on the fracture toughness in sand concrete.

In order to study the effect of fine aggregate characteristics on the fracture toughness of sand concrete, aggregates with different fineness modulus, FAA and aggregate gradation were selected as raw materials. The brittleness coefficient of sand concrete specimens with different fine aggregates was measured and compared with traditional concrete. Moreover, the complexity of the fracture surface and the evolution of the microstructures of sand concrete specimens were analyzed by means of box-counting fractal dimensions and Scanning Electron Microscope (SEM), respectively. This research will contribute to a better understanding of how the characteristics of fine aggregate affect the fracture toughness of sand concrete [23,24,25].

## 2. Experimental Programs

### 2.1. Experimental Materials

#### 2.1.1. Fine Aggregate

Fine aggregates here are defined as grain sizes smaller than 2.36 mm. The used fine aggregates are gold tailing sand (GTS) from Tong-guan county, iron tailing sand (ITS) from Shangluo City and river sand (RS) from Xi’an City (shown in Figure 2). The results indicate that GTS is finer and the RS is coarser. And the fine aggregate with different particle size distributions shows different colors.

#### 2.1.2. Cement

Ordinary Silicate Cement, 42.5 grade of conch brand, was used in experiments. The physical characteristics of the cement used are a specific density: 1030 ± 20 kg/m^3^ and a specific surface area: 446.5 m^2^/kg.

#### 2.1.3. Admixtures

The admixture used is a superplasticizer with 28% of dry matter content in liquid form, and the water-reducing capacity was greater than 25%.

### 2.2. Mix Proportion and Specimen Preparation

The mix proportions for specimen preparation in this study were summarized in Table 1. The fluidity was adjusted and controlled at 220 ± 5 mm by using a superplasticizer in the experiment (with a content of 3.1~3.6% of cement mass). It should be noted that as the fineness of the fine aggregate decreases, a more water-reducing agent is required, but since the amount of water-reducing agent is relatively low, the impact on the cost of concrete is not significant. In order to keep the fluidity consistent, there is a certain increase in water demand with the cement aggregate ratios going from 1:3 to 1:1 because a mass of fine powder led to the higher surficial area of aggregate, which demanded more water to lubricate those fine particles. The weighted cement and the fine aggregate were added into the mixer in turn and dry-mixed for 1 min at a slow speed; then, the superplasticizer and water were mixed thoroughly and poured into the mixer for 3 min at a slow speed. Finally, the mixtures were also mixed for 3 min at high speed to ensure homogenization. After tested flowability, the mixture was poured into molds to make 40 mm × 40 mm × 160 mm specimens, placed for 24 h and then de-molded. After de-molding, the specimens were moved to the standard maintenance room maintenance for 7 and 28 days (curing temperature of 20 ± 2 °C, curing humidity ≥ 95%).

### 2.3. Test Methods

#### 2.3.1. Physical Properties Test of Fine Aggregates

In order to analyze the effect of particle characteristics of fine aggregates on the toughness of sand concrete, the physical properties of three kinds of fine aggregates were conducted. The size distributions, fineness modulus, apparent density, bulk density, void ratio and powder content of the fine aggregates were tested following the national standards of GB/T 14684-2011 (Standard of Sand for Construction) used in China. And the FAA determination device required by the cited JTD E42-2005 standard.

#### 2.3.2. XRD Test

X-ray Diffraction (XRD) analyses were performed to detect the phase compositions of fine aggregate. Firstly, fine aggregates were grounded into powder, passed through 200 mesh sieves, and then dried for 24 h in a drying oven at 105 °C. Finally, characterized the composition and content of fine aggregate using a D8 ADVANCE X-Ray Diffractometer. The measurement conditions of XRD were a tube voltage of 30 kV and a tube current of 20 mA. Scans were repeated continuously from 15° to 80° with 0.1°/s using Cu target (wavelength is 1.54178 Å).

#### 2.3.3. Compressive Strength and Flexural Strength Test

According to the strength test of cement mortar with GB/T 17617-2005. The specimens were tested for compressive strength and flexural strength at the curing periods of 7 and 28 days by means of using the DYE-300 automatic universal testing machine. The loading rate is 2400 N/s, and the flexural loading rate is 50 N/s. The sizes of the specimens were all 40 mm × 40 mm × 160 mm.

#### 2.3.4. Box-Counting Fractal Dimensions Calculation

The complexity of the concrete fracture surface can be characterized by the box-counting fractal dimension [26,27]. The fracture energy of concrete increases with the complexity of the fracture surface. Accordingly, the fracture toughness of concrete increases with the higher box-counting fractal dimension. Hence, the box-counting fractal dimensions of the fracture surface with the SC1 and OC1 specimens were calculated by using MATLAB. And the fracture toughness of the concrete is characterized by box-counting fractal dimension.

#### 2.3.5. SEM Test

The macroscopic mechanical properties of sand concrete are closely related to the microstructure of sand concrete. So, the microstructure of the specimens of SC1 and OC1 were tested by Hitachi S-4800 Scanning Electron Microscope. The specimens were first immersed in absolute ethanol solution for 24 h in order to terminate the hydration reaction. Regular pieces of a 5-mm size were cut, polished and then kept in an oven for 24 h at 105 ± 5 °C. Finally, the microstructures of the specimens were tested after gold coating.

## 3. Results and Discussion

### 3.1. The Characteristics of Fine Aggregate

The mineral compositions of the fine aggregates are shown in Figure 3. The main minerals of different fine aggregates are quartz and feldspar, which are silicate minerals with stable properties and can be used as aggregates. Results also indicate that the mineral composition and content of GTS, ITS and RS are close. And quartz content up to 70%. The chemical compositions of the fine aggregates are shown in Table 2. The main compositions of GTS, ITS and RS are SiO_2_, Al_2_O_3_ and Fe_2_O_3_. The river sand is constantly washed in water, and its mineral composition is relatively single; while tailings sand exists in the field environment, its mineral composition is more complex. Therefore, the quartz content of river sand is higher than that of quartz in tailings sand.

The size distributions of the fine aggregates are shown in Figure 4. The results showed that particle sizes of GTS range from 0.02 to 1.18 mm, and the particle sizes of ITS and RS range from 0.02 to 2.36 mm. The fine aggregate presents a continuous particle size distribution. But the particle size distribution of RS is slightly more spread out than that of GTS and ITS. The sets of physical characteristics for different aggregates are listed in Table 3. This table reveals that the density of ITS is slightly higher than GTS and RS. The fineness modulus of fine aggregates is 0.62 for GTS, 1.29 for ITS and 1.92 for RS, while the void ratio of fine aggregates is 47.62% for GTS, 45.33% for ITS and 43.57% for RS, respectively. It can be inferred that as the fineness modulus of fine aggregates decreases, the void ratio of fine aggregates increases. The powder content (<0.075 mm) also varied among the different fine aggregates, with GTS accounting for 32.91%, ITS for 16.73%, and RS for 6.92%. It should be noted that these proportions of fine powder are still acceptable in sand concrete, as the fine powder may promote the heterogeneous nucleation in fresh concrete, which will indirectly improve the hydration process and the microstructure of the cement paste in a positive way [28].

The micromorphology of the fine aggregates is shown in Figure 5. The results showed that GTS grains (Figure 5a) and ITS grains (Figure 5b) are angular shapes, while RS grains are round shapes (Figure 5c). The FAA of different fine aggregates was tested to analyze the effect of shape characteristics on the property of sand concrete. Five replicates of each fine aggregate were measured according to the angularity experiment of fine aggregate in the JTG E42-2005 engineering test procedure, and the average FAA values are shown in Figure 5. The test results showed that the FAA values were 39 s for GTS, 44 s for ITS and 32 s for RS.

The gradation characteristics of different fine aggregates also affect the fracture toughness of sand concrete. Hence, in this study, the modified Andreasen close packing model [29] is utilized to compare the gradation quality of different fine aggregates, which is shown as follows:(1)$$PD=\frac{D^{q}-D_{min}^{q}}{D_{max}^{q}-D_{min}^{q}}$$
where *D* is the particle size (mm), *P(D)* is a fraction of the total solids being smaller than size *D*, *D_min_* is the minimum particle size (mm), *D_max_* is the maximum particle size (mm) and *q* is the distribution modulus. The value of *q* is between 0.23 and 0.37. The *q* value of the target curve in the close packing model is fixed at 0.23 based on the properties of fine aggregates [29]. The gradation quality of the fine aggregate is assessed by the determination coefficient of the target curve (*R*^2^, as shown in Equation (2)).
(2)$$R^{2}=1-\frac{\sum_{i=1}^{n}{\left(P_{mix}\left(D_{i}^{i+1}\right)-P_{tar}(D_{i}^{i+1})\right)}^{2}}{\sum_{i=1}^{n}{\left(P_{mix}\left(D_{i}^{i+1}-\bar{P_{mix}}\right)\right)}^{2}}$$
where *P_mix_* is mixed fine aggregates, the *P_tar_* is the target grading calculated from Equation (1), *n* is the number of points used to calculate the deviation and $\bar{P_{mix}}$ represents the mean of the entire distribution. The target curve under this model and the particle size distribution curve of different fine aggregates are shown in Figure 6. The results showed that the correlation coefficient of ITS (*R*^2^ = 0.86) was highest compared to the target curve, while the correlation coefficient of GTS(*R*^2^ = 0.64) and RS (*R*^2^ = 0.60) were lower compared to the target curve. As a result, ITS showed the most reasonable aggregate gradation, while RS showed the most undesirable aggregate gradation. This suggests that the gradation is closely related to the particle size distribution of fine aggregate. Finally, it should be noted that the basic difference between these aggregates lies in physical characteristics, grain shape, granularity and particle size distributions.

### 3.2. The Brittleness Coefficient of Sand Concrete with Different Fine Aggregates

The results of the compressive strength and flexural strength of sand concrete with different fine aggregates are shown in Figure 7. Based on the obtained results, it can be concluded that the compressive strengths and flexural strengths of sand concrete increase with the increase of the cement aggregate ratios and then basically remain constant. The compressive strength and flexural strength of ITS sand concrete are higher than that of GTS concrete and RS concrete at different cement aggregate ratios. This indicates that the mechanical strength of sand concrete is closely related to gradation and FAA. Since the gradation of ITS was more reasonable than that of GTS and RS, which promoted the cohesion of concrete. Meanwhile, higher FAA means more contact among fine aggregates, which resists more compaction, and the mechanical properties are improved.

The brittleness coefficient (ratio of compressive strength to flexural strength) reflects the deformation resistance of concrete and can characterize the toughness of concrete simply and intuitively. The lower the brittleness coefficient, the better the resistance to deformation. According to the GB/T50010-2010 code for the design of concrete structures, the brittleness coefficient of ordinary concrete is between 8:1 and 11:1, and it becomes higher with the increase of concrete strength grade. Figure 8 shows that the brittleness coefficient of sand concrete with different aggregates ranged from 5:1 to 6.5:1; however, the brittleness coefficient of fiber-reinforced concrete ranged from 5:1 to 6:1 [30,31], which is similar to that of fiber reinforced concrete, and the brittleness coefficient of sand concrete changes more slowly than ordinary concrete as the strength grade increases. This indicates that the resistance to deformation and fracture toughness of sand concrete is better than ordinary concrete. The results also show that the brittleness coefficient of sand concrete with ITS is the lowest and the highest with RS, which implies that the brittleness coefficient of sand concrete decreases and the fracture toughness increase as the fine aggregate gradation become better and the FAA increases.

### 3.3. Box-Counting Fractal Dimensions of Fracture Surface

The box-counting fractal dimensions of different concrete fracture surfaces are illustrated in Figure 9. The fracture surface of SC1 and OC1 specimens with a curing time of 28 days were images binarized using MATLAB to further analyze the reason for the low brittleness coefficient of sand concrete. Finally, the box-counting fractal dimensions of the binarization image were calculated using MATLAB to characterize the complexity of the concrete morphology of the fracture surface [32]. The absolute value of the slope of the linear equation in the figures is the fractal dimension. The results show that the fractal dimensions are 2.69 for ITS sand concrete, 2.45 for GTS sand concrete, 2.25 for RS sand concrete and 2.11 for ordinary concrete with similar compressive strength. This indicates that the fractal dimension of sand concrete is higher than that of ordinary concrete, and the fracture surface morphology of sand concrete is more complex than that of ordinary concrete because sand concrete not only has no coarse aggregates, but the fine aggregates used have finer modulus, higher complexity and roughness than ordinary concrete [33]. Therefore, more energy was consumed during the process of fracture in sand concrete and showed higher fracture toughness. The results also show that the box-counting fractal dimensions of the ITS sand concrete surface and GTS sand concrete surface are higher than those of RS sand concrete since the particles of tailings are sharp angles, while the river sand particles are smooth and round. The angularities of ITS are 9.3% higher than that of GTS and 25.6% higher than that of RS. Therefore, the complexity and roughness of the fracture surface of the ITS sand concrete specimen are the highest, while that of the RS sand concrete specimen is the lowest. This suggested that the angularity of the aggregate affected the complexity and roughness of the concrete fracture surface. The complexity of concrete fracture surfaces and the fracture toughness of concrete increases with the FAA of aggregate increasing.

### 3.4. Microstructural Analysis

The microstructure and properties of ITZ are one of the most important influencing factors on the mechanical properties of concrete [34,35,36]. The weak region of ITZ is prominent and relevant to the overall mechanical strength; moreover, it can provide an easier path for crack propagation [37]. The microstructure of the SC1 and OC1 at 28 days were characterized by SEM to further analyze the effects of microcrack and hydration products on the fracture toughness of sand concrete.

The propagation paths of microcracks in different sand concrete and ordinary concrete are shown in Figure 10. The microcracks started to develop in the mortar matrix and then developed around the aggregates. It can be seen from Figure 10d that the microcracks in ordinary concrete grew along the coarse aggregate interface, and less energy was consumed during crack propagation of the specimen due to little crack deflection. The direction and path of crack propagation in sand concrete (Figure 10a–c) are slightly different, but there is a common point that the fracture surface showed an irregular, tortuous shape along the fine aggregate, and the path of crack deflection is complex, which can translate to higher energy dissipation capability. Since fine aggregates are very hard, cracks will only deflect or branch along the fine aggregate particles and form deflection cracks in the matrix. The paths of crack propagation were prolonged in the matrix, and more fracture energy was consumed under the bending load. This further confirms that the fracture toughness of sand concrete can be improved by means of crack deflection [38]. It is deserved attention that tailing sand concrete forms more crack deflections than river sand concrete because the tailing sand is more sharply angled than river sand. This shows that the FAA affects the fracture toughness of sand concrete, and the angular aggregates in sand concrete resist the growth of microcracks better than round aggregates.

The adhesion between the aggregate and the cement paste also affects the fracture toughness of the concrete. The widths of microcracks in the ITZ between the aggregate and cement paste in sand concrete with different fine aggregates and ordinary concrete are shown in Figure 11. The results show that the adhesion of tailings sand and paste is tight in the ITS sand concrete (Figure 11a) and GTS sand concrete (Figure 11b); there are narrower microcracks in the ITZ, which are 1.26 μm for the ITS sand concrete and 1.48 μm for the GTS sand concrete respectively. While the adhesion between river sand and paste is loose in the RS sand concrete (Figure 11c) and ordinary concrete (Figure 11d), and there are wider microcracks in the ITZ, which are 2.05 μm for the RS sand concrete and 3.14 μm for ordinary concrete respectively, which implies the weaker bonding effect between the surfaces of river sands and harden paste due to the round and slick surfaces of river sands. The characteristics of the tailings sand provided by its angularity and rough surface contribute to stronger adhesion, which will facilitate its bonding with the main hydration products; this means that a better mechanical interlock between the aggregate and the cement matrix can be achieved, therefore producing finer microcracks. Figure 11a,b also show that the calcium silicate hydrates (C-S-H) phases cross the cracks in the tailings sand concrete. Due to the bridging action, the energy-absorbing effect can inhibit further propagation of cracking and crack widening. This refined microstructure could be attributed to the role of good gradation tailing sand in regulating the shape and assembly of hydration products.

The hydration products of sand concrete with different aggregates and ordinary concrete are shown in Figure 12. The main hydration products are compact C-S-H phases with small amounts of hexagonal plate calcium hydroxide (CH) crystals and needle-like structures of ettringite(AFt) (Figure 12a,b), while the hydration products are C-S-H phase with a large number of hexagonal plate CH crystals and needle-like shape or rod-like AFt in river sand concrete (Figure 12c) and ordinary concrete (Figure 12d). Because the correlation coefficient between the size distribution curves of tailings sand and the target curve is higher than that between the size distribution curves of the river sand and the target curve under the Andreasen close packing model, this suggests that the compactness and gradation of tailings sands are better than river sands, which significantly reduce the free space between the fine aggregate and cement paste, so the lack of free space restrains the full growth of crystals. Most small crystals are embedded in the dense C-S-H gel bulk and are difficult to observe. Consequently, the ITZ of tailings sand concrete is more compact than the ITZ of RS sand concrete. The reduction of CH and AFt content in sand concrete not only improves the stability of the structure but also effectively improves the weakness of the ITZ in sand concrete, which has a positive effect on improving the fracture performance of sand concrete. The content of the powder filler also has an important effect on the properties of sand concrete; the mechanical properties will decrease beyond a certain range [39]. The ITZ of GTS concrete is weaker due to the high powder content, as the powder content of GTS reaches 32.91%, while the powder content of ITS is 16.93%. This suggests that the weakness of the ITZ in sand concrete is affected by the fineness and gradation of fine aggregate. Figure 12 also shows that the CH content of ITS sand concrete is lower than GTS sand concrete. On the one hand, the particle size distribution of ITS fine aggregates is wider and has better gradation. On the other hand, the powder content of GTS fine aggregate is higher than that of ITS fine aggregates, which increases the void ratio between the fine aggregates and provides more space for the growth of CH crystals and AFt. Due to the poor microstructure of ITZ, the brittleness coefficient of ordinary concrete is much higher than that of sand concrete. Therefore, the different characteristics of the fine aggregates lead to different hydration products in the sand concrete, which affect the fracture toughness of the sand concrete.

## 4. Conclusions

From the data of experimental in this study, the following conclusions can be drawn:(1)The fracture toughness of sand concrete is related to the characteristics of the aggregate. However, the brittleness coefficients of sand concrete with different aggregates are between 5:1 and 6.5:1 at different strength grades, and the brittleness coefficient of sand concrete is much lower than that of ordinary concrete. Sand concrete exhibits good fracture toughness than ordinary concrete.(2)The effect of particle fineness of fine aggregates on the fracture toughness of sand concrete is not significant.(3)The fracture toughness of sand concrete is strongly affected by the FAA. As the FAA value increases from 32 s to 44 s, the fractal dimension of the fracture surface of sand concrete increases from 2.25 to 2.69, and the average flexural strength increases by 13.65% under the same compressive strength.(4)The fracture toughness, microstructure and hydration products of sand concrete are also related to the gradation of fine aggregates. A good grading distribution can improve the performance of ITZ. When the correlation coefficient of the gradation compared to the target curve increased from 0.60 to 0.86, the microcrack width of ITZ decreased by about 38.55%, and there was also a difference in the hydration products produced. Therefore, the better the gradation of fine aggregates, the better the crack resistance of sand concrete.

## Figures and Tables

**Figure 1 materials-16-02080-f001:**
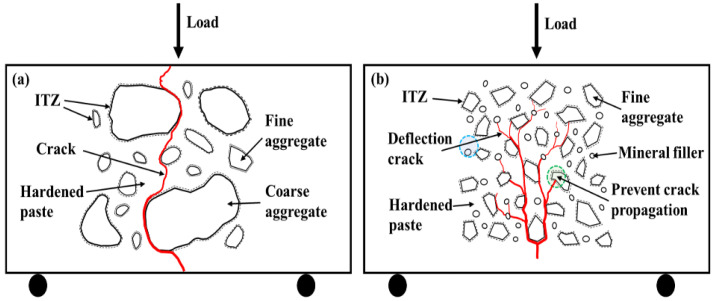
Crack propagation of sand concrete. (**a**) sand concrete with round and coarse aggregate, (**b**) sand concrete with angular and fine aggregate.

**Figure 2 materials-16-02080-f002:**
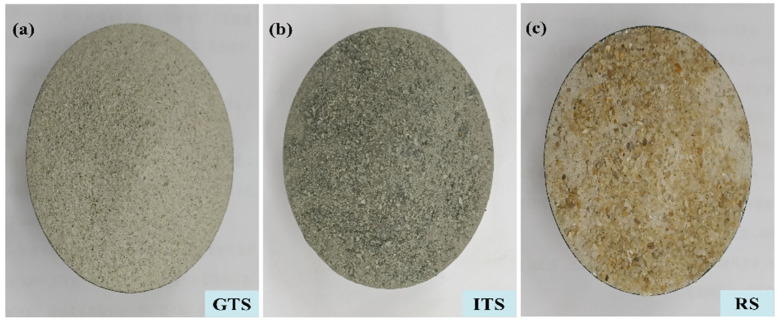
Samples of three fine aggregates. (**a**) GTS, (**b**) ITS, (**c**) RS.

**Figure 3 materials-16-02080-f003:**
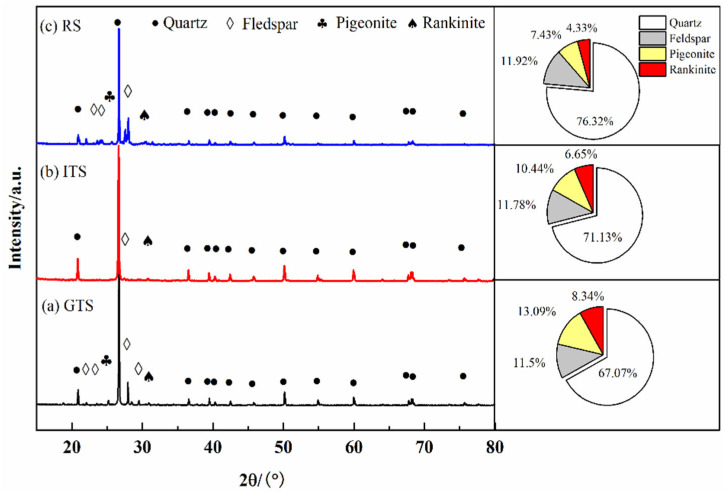
Mineral component and content of aggregate (**a**) GTS, (**b**) ITS and (**c**) RS.

**Figure 4 materials-16-02080-f004:**
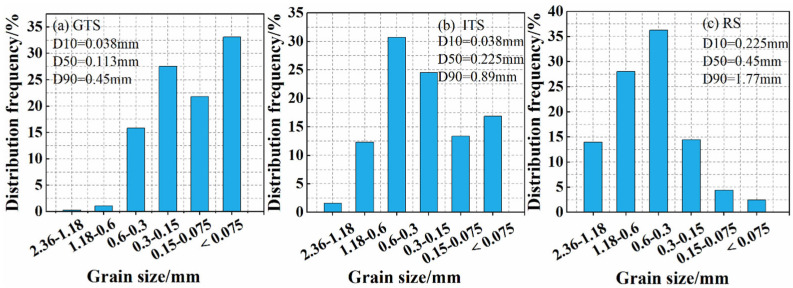
Size distributions of different fine aggregates (**a**) GTS, (**b**) ITS, (**c**) RS.

**Figure 5 materials-16-02080-f005:**
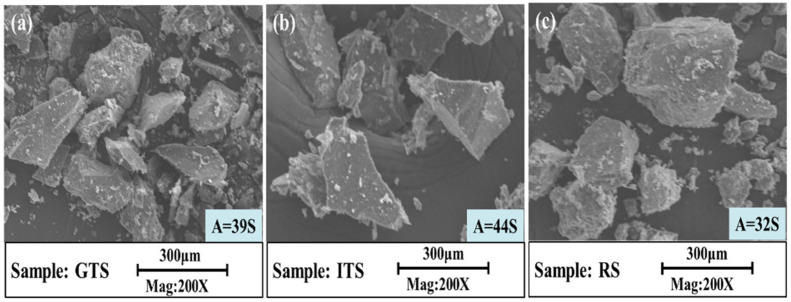
Microstructure and angularity of fine aggregates (**a**) GTS, (**b**) ITS, (**c**) RS.

**Figure 6 materials-16-02080-f006:**
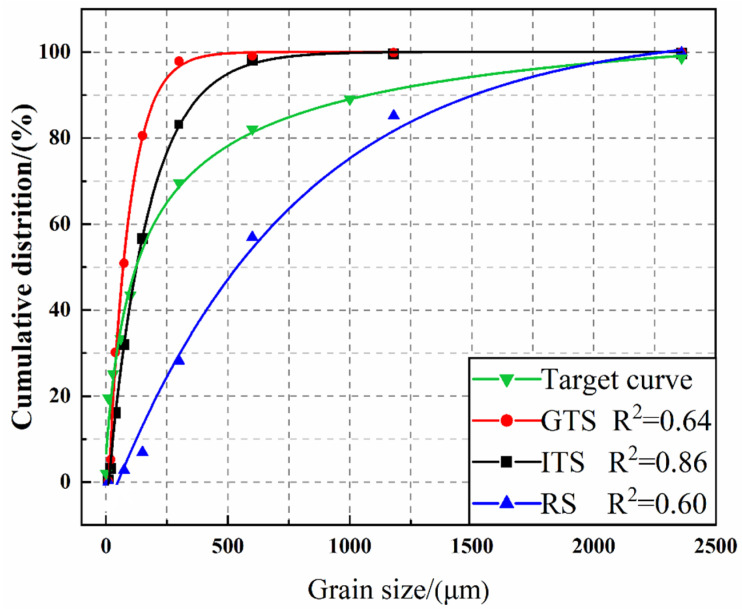
Size distribution curves of fine aggregate and target curves under Andreasen close packing model.

**Figure 7 materials-16-02080-f007:**
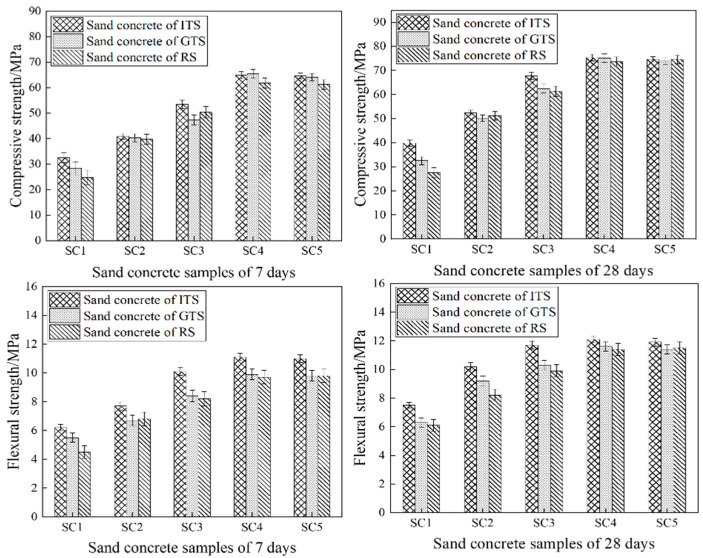
Compressive strength and flexural strength of sand concrete.

**Figure 8 materials-16-02080-f008:**
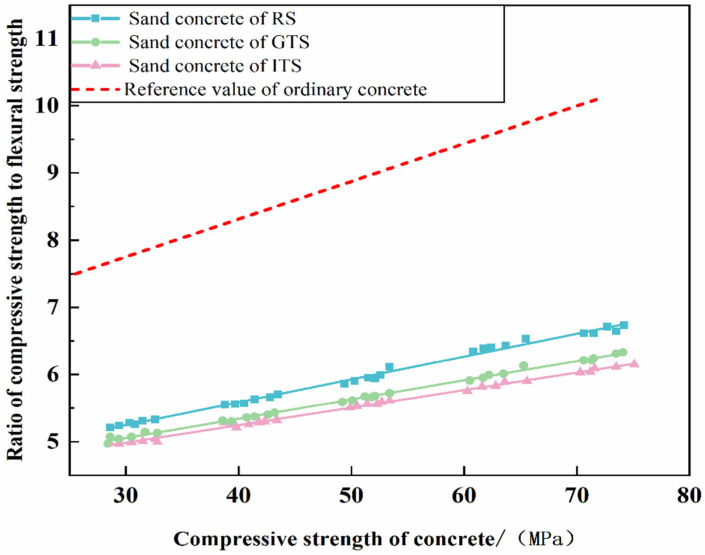
The ratio of compressive strength to flexural strength with sand concrete.

**Figure 9 materials-16-02080-f009:**
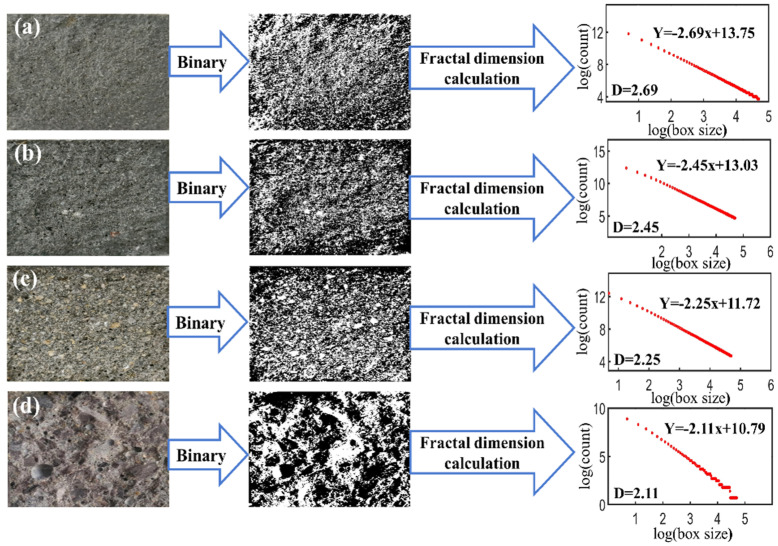
Binarization of image and fractal dimension calculation (**a**) ITS sand concrete, (**b**) GTS sand concrete, (**c**) RS sand concrete, (**d**) Ordinary concrete.

**Figure 10 materials-16-02080-f010:**
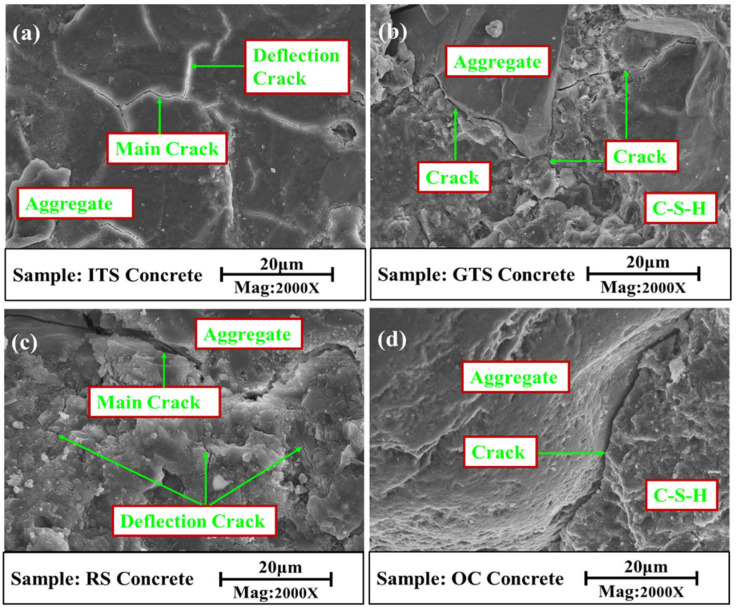
Propagation paths of microcracks in concrete (**a**) ITS sand concrete, (**b**) GTS sand concrete, (**c**) RS sand concrete, (**d**) Ordinary concrete.

**Figure 11 materials-16-02080-f011:**
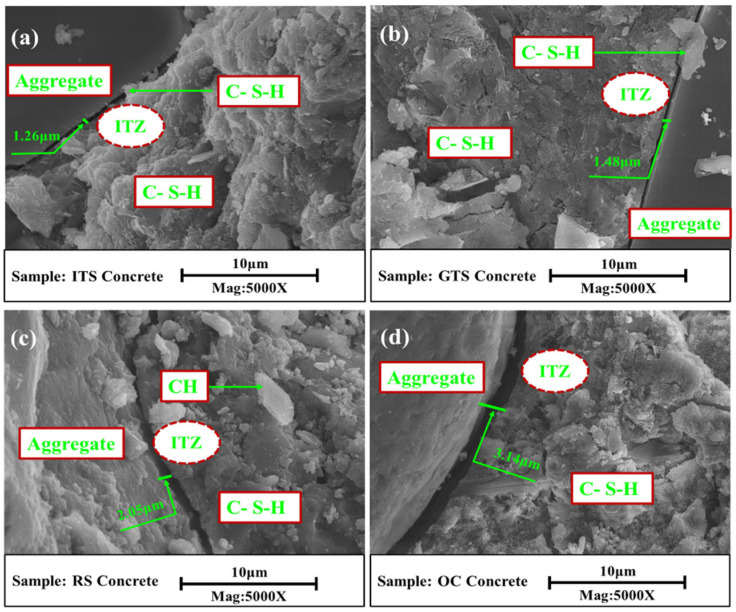
ITZ of concrete (**a**) ITS sand concrete, (**b**) GTS sand concrete, (**c**) RS sand concrete, (**d**) Ordinary concrete.

**Figure 12 materials-16-02080-f012:**
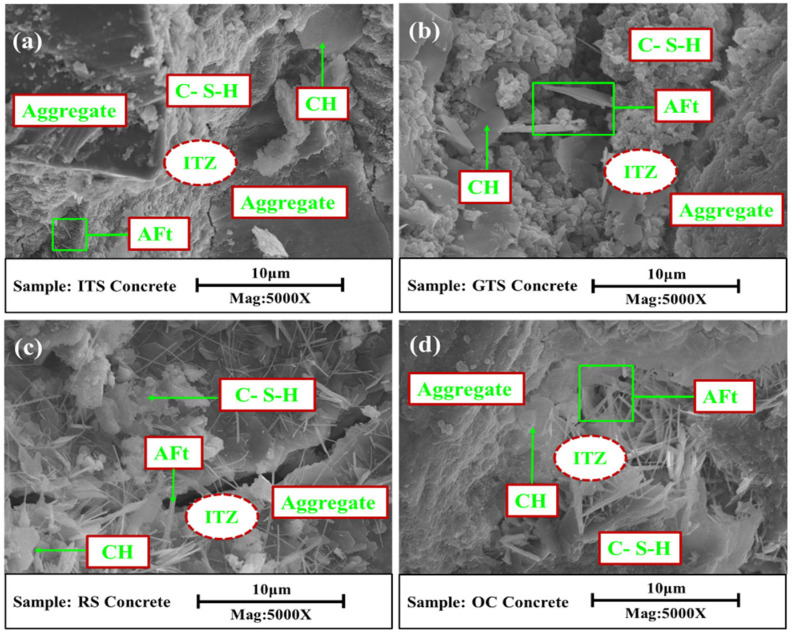
Hydration products of concrete (**a**) ITS sand concrete, (**b**) GTS sand concrete, (**c**) RS sand concrete, (**d**) Ordinary concrete.

**Table 1 materials-16-02080-t001:** Mix proportions of materials used to produce sand concrete.

Specimens	Ratio of Cement to Sand	Cement /kg·m^−3^	GTS /kg·m^−3^	ITS /kg·m^−3^	RS /kg·m^−3^	Water /kg·m^−3^
SC1		585.9	—	1757.1	—	310.5
	1:3	585.9	1757.1	—	—	310.5
		585.9	—	—	1757.1	310.5
SC2		669.3	—	1673.7	—	274.4
	1:2.5	669.3	1673.7	—	—	274.4
		669.3	—	—	1673.7	274.4
SC3		781.2	—	1561.8	—	234.4
	1:2	781.2	1561.8	—	—	234.4
		781.2	—	—	1561.8	234.4
SC4		937.5	—	1405.5	—	206.3
	1:1.5	937.5	1405.5	—	—	206.3
		937.5	—	—	1405.5	206.3
SC5		1171.5	—	1171.5	—	175.7
	1:1	1171.5	1171.5	—	—	175.7
		1171.5	—	—	1171.5	175.7

Note: The control specimen of OC1 is designed with a strength grade of C30 (The ratio of cement, sand, pebbles and water is 1:1.11:2.72:0.38 s).

**Table 2 materials-16-02080-t002:** Chemical composition of fine aggregate.

Type	Chemical Compositions of Raw Materials (wt. %)											
	SiO_2_	Al_2_O_3_	Fe_2_O_3_	CaO	MgO	Na_2_O	K_2_O	P_2_O_5_	TiO_2_	MnO	LOI	TOTAL
GTS	73.34	7.06	3.82	3.22	1.26	1.58	2.08	0.25	0.34	0.10	3.66	96.71
ITS	75.85	7.54	6.64	4.19	2.12	0.60	0.48	0.13	0.30	0.15	0.86	98.86
RS	81.32	4.75	2.83	2.50	0.48	0.23	1.57	0.09	0.18	0.05	2.16	96.16

**Table 3 materials-16-02080-t003:** Physical characteristics of fine aggregate.

Type	Fineness Modulus	Apparent Density (g/cm^3^)	Bulk Density (g/cm^3^)	Void Ratio (%)	Average Size (µm)	Specific Area (m^2^/kg)	Powder Content (<0.075 mm)
GTS	0.62	2.67	1.40	47.62	113	323	32.91%
ITS	1.29	2.78	1.52	45.33	225	282	16.93%
RS	1.92	2.64	1.48	43.57	450	256	6.92%

## Data Availability

Not applicable.

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
