# Peer review of "Effect of Different Fine Aggregate Characteristics on Fracture Toughness and Microstructure of Sand Concrete"

_materials, 2023, doi:10.3390/ma16052080_

Round 1

Reviewer 1 Report

This study investigates the influence of fine aggregate characteristics on sand concrete's fractal dimension and microstructure. Three different fine aggregates were used in this study (gold tailing sand, iron tailing sand and river sand) to evaluate the compressive strength, flexural strength, fractal dimension and microstructure characteristics of sand concrete. This article needs significant improvement before it can be accepted for publication.

1. Abstract: the current problem in sand concrete and their fracture mechanics, the purpose of the study should be improved. The current version is general and shallow.

2.   The work described in the abstract is shallow and should be in detail. Additionally, the quantified results could be included rather than general findings.

3. The fracture toughness of concrete beams is usually evaluated using the area under the load-deflection curve. Some researchers used small-scale cylindrical or rectangular specimens with a notch to evaluate the fracture toughness under mode I, II, III and mixed modes. In this study box-counting fractal dimensions test was used to evaluate fracture toughness. Is it fracture toughness or fracture surface of concrete?. Please clarify this.

4. Figure 6, how to ensure the accuracy of the cumulative distribution for the GTS?. The R2 value for GTS is 0.64, which is an unacceptable limit. What is the significance of R2 values presented in Figure 6?

5. What is the maximum dosage of water-reducing admixture?. The 28% used in this study is so high. Please check

6. Please remove section 2.1.4.

7. Table 1, on what basis the ratio of cement to sand was selected?

8. Section 2.3.2. please mention the sample size used for the compressive and flexural strength tests.

9. Include the picture of the Box-counting fractal dimensions test setup.

10. Why is the quart content higher in RS than ITS and GTS?. Please highlight the reason for this behaviour in section 3.1.

11. Section 3.2; compare the results or trend with the literatures.

12. Section 3.3: there is no discussion about fracture toughness? Most discussions was made about the fractal dimension. Please clarify this

13. Conclusions should be improved. Please add more key findings.

Reviewer 2 Report

The work submitted to the Sustainability journal titled as “Effect of different fine aggregate characteristics on fracture toughness and microstructure of sand concrete” is reviewed. In general, whole manuscript requires minor editing. The reviewer also thinks that this is a very interesting research and supported with the microscope investigation. Findings are accurate and consistent with the mechanical properties. However, the reviewer thinks that introduction section should be revised to express the novelty of this research. I am suggesting a revision if the authors are willing to perform improvements / corrections on the submitted work as mentioned below;

-        Abstract, line 19; what do the authors mean by “more reasonable”?

-        Abstract, lines 20-21; typo error “these results demonstrate”.

-        P.1, lines 42-43; it would be much clear if authors specify up to which percent fracture toughness of sand concrete increased. Same applies to line 47, please define “later stage”.

-        P.2, line 72; the reviewer believes that the sentence needs revision as “are also influence by the aggregate gradation” as this is one of the factors.

-        Section 1; In general, introduction section focusses on the use of mineral admixtures and their effects on the concrete performance. However, it is not the main aim of this study to review mineral admixture incorporated concretes. The reviewer thinks that this section should be the main section to review earlier relevant studies and link it with the aim of the study. The emphasis may be given to the use of varying types of sands. Also, novelty of this research should be highlighted. The reviewer thinks that author may consider revising introduction section to fulfil above-mentioned point.

-        P.3, section 2.1.1.; the reviewer thinks that it would be also appropriate to include particle size distribution (in log graph) of the sand types used. This would support grading related effects.

-        P.3, section 2.1.3; this section requires re-writing.

-        P.3-4; Chemical admixture content required for the same consistency should also be stated. This would be important for the scientific community if cost assessment is carried out for the produced concrete.

-        Figure 7; y-axis should be the same for 7 & 28 days for both compressive and flexural strength values. Even though, strength development is not the concern for this research, the reviewer thinks that it would be more appropriate.

-        It is clear from the manuscript that as the angularity increases, it would require higher energy during the fracture. However, angular shaped materials are not mostly demanding due to their angular characteristics increase internal friction and then have adverse effect on the fresh properties. As the fineness modulus increase, having angular shaped particles would increase water demand substantially. This would increase the chemical admixture to compensate water absorption of the mix and thereby it could potentially increase the cost of the mix. The reviewers kindly ask authors to include comments on this matter and include it in the context.

Reviewer 3 Report

The text that is the subject of this review is devoted to analysing the impact of selected properties of the aggregates used to prepare sand concrete on the compressive strength of the finished product. This topic shows excellent scientific and application potential, so undertaking such research is reasonable.

The obtained results are valuable and may allow the development of science and technology, particularly in building materials, including sand concrete.

While carefully reading the text, the following observations come to mind:

  • The caption of Figure 2 is "macroscopic morphology of fine aggregates particle - The image shows only relatively low-resolution photos of sands that cannot be called morphology - there is no information on the quantitative or even qualitative features of this morphology.
  • Table 1 shows the compositions of specimens SC1-SC5, and below it is stated that the "control group" (control specimen?) "of OC1 was designed with a strength grade of C30". Was the OC1 sample made with the same cement as SC1-SC5? What sand and gravel were used to compose this sample? In the further text, the details of the morphology of the samples obtained from the GTS, ITS and RS sands were considered - it seems that the composition of the OC1 sample will be necessary, especially since the morphology of the obtained concretes is considered in the text.
  • Is the OC1 mentioned in L122 the same specimen as "common concrete" in Figure 8 or "ordinary concrete" in Figures 9-12?
  • In L142 and L 147 is written that the specimens SC1 and OC1 were tested (using particular apparatus) – only these specimens were tested? what about SC2, SC3, SC4 and SC5? In the further text, the results of tests for SC2-SC5 were also shown.
  • L164-168 and figure 4 show the size distributions of fine aggregates. What was the measuring device used to determine this characteristic? It is not described in chapter 2.3.
  • In addition to the histogram in figure 4, wouldn't it be worth giving quantitative indicators like percentiles: d50, d90 and d10?
  • In table 3, values of physical characteristics are shown. What was the measuring device used to determine this characteristic? It is not described in chapter 2.3
  • What device was used for the measurement of angularity shown in figure5? The FAA determination device required by the cited JTD E42-2005 standard is not shown in Chapter 2.3.
  • Equation (2) - L201-202 is signed twice.
  • In L240, "gradation improved" is written, which quantitatively means this "improvement" - increasing or decreasing gradation?
  • What is presented in Figure 9? What is the difference between photos with suffix 1 and those with suffix 2?
  • What are the X and Y values in the following equations in these graphs? What is this "linear equation"? Information on the methodology of determining the fractal dimension, especially in the sentence from L254, is lapidary and should be expanded.
  • In figure 9, the axes are not described - what are the values marked in the following graphs in figure 9?
  • The unit "MPa" is in figures 7 and 8 written as "Mpa" – it should be corrected.

The indicated imperfections do not diminish the text's substantive value but hinder its reception and should be corrected following the suggestions.

Round 2

Reviewer 1 Report

All comments are addressed by the author adequately.